# The Prophylactic Effects of Glutamine on Muscle Protein Synthesis and Degradation in Rats with Ethanol-Induced Liver Damage

**DOI:** 10.3390/nu13082788

**Published:** 2021-08-14

**Authors:** Qian Xiao, Yi-Hsiu Chen, Satwika Arya Pratama, Ya-Ling Chen, Hitoshi Shirakawa, Hsiang-Chi Peng, Suh-Ching Yang

**Affiliations:** 1School of Nutrition and Health Sciences, Taipei Medical University, Taipei 11031, Taiwan; doyouknow19950611@gmail.com (Q.X.); ma07108007@tmu.edu.tw (Y.-H.C.); pratama94@gmail.com (S.A.P.); ylchen01@tmu.edu.tw (Y.-L.C.); hcpeng@tmu.edu.tw (H.-C.P.); 2Nutrition Department, State University of Surabaya, Surabaya 60213, Indonesia; 3Laboratory of Nutrition, Graduate School of Agricultural Science, Tohoku University, Sendai 980-8857, Japan; shirakah@m.tohoku.ac.jp; 4Research Center of Geriatric Nutrition, College of Nutrition, Taipei Medical University, Taipei 11031, Taiwan; 5Graduate Institute of Metabolism and Obesity Sciences, Taipei Medical University, Taipei 11031, Taiwan; 6School of Gerontology Health Management, College of Nursing, Taipei Medical University, Taipei 11031, Taiwan; 7Nutrition Research Center, Taipei Medical University Hospital, Taipei 11031, Taiwan

**Keywords:** glutamine, muscle protein synthesis, muscle protein degradation, gut permeability, microbiota composition, ethanol-induced liver injury, Wistar rats

## Abstract

The purpose of this research was to investigate the prophylactic effects of glutamine on muscle protein synthesis and degradation in rats with ethanol-induced liver injury. For the first 2 weeks, Wistar rats were divided into two groups and fed a control (*n* = 16) or glutamine-containing diet (*n* = 24). For the following 6 weeks, rats fed the control diet were further divided into two groups (*n* = 8 per group) according to whether their diet contained no ethanol (CC) or did contain ethanol (CE). Rats fed the glutamine-containing diet were also further divided into three groups (*n* = 8 per group), including a GG group (glutamine-containing diet without ethanol), GE group (control diet with ethanol), and GEG group (glutamine-containing diet with ethanol). After 6 weeks, results showed that hepatic fatty change, inflammation, altered liver function, and hyperammonemia had occurred in the CE group, but these were attenuated in the GE and GEG groups. Elevated intestinal permeability and a higher plasma endotoxin level were observed in the CE group, but both were lower in the GE and GEG groups. The level of a protein synthesis marker (p70S6K) was reduced in the CE group but was higher in both the GE and GEG groups. In conclusion, glutamine supplementation might elevate muscle protein synthesis by improving intestinal health and ameliorating liver damage in rats with chronic ethanol intake.

## 1. Introduction

Alcoholic liver disease (ALD) is the top-ranked major disease burden worldwide [1]. Furthermore, chronic liver disease and cirrhosis are two of the top ten leading causes of death in Taiwan [2]. Therefore, ALD has become a serious problem that needs to be addressed. Sarcopenia, as one of the major symptoms of malnutrition in liver diseases, is commonly found in ALD patients. Around 60% of ALD patients suffer various extents of loss of muscle mass [3]. Muscle loss in ALD is mainly caused by malnutrition due to liver dysfunction [4,5]. On the other hand, some studies also indicated that chronic alcohol intake also induces intestinal leakage and alters the gut microbiotic composition, which allows bacterial translocation into the body, producing endotoxemia and thereby dysregulating factors involved in protein synthesis, degradation, or autophagy in muscles [6,7]. Branched-chain amino acids (BCAAs), which are commonly used to increase muscle protein synthesis, cannot inhibit muscle loss in cirrhotic patients, because of a defect in amino acid metabolism due to reduced liver function [5]. New approaches are needed to ameliorate muscle loss in ALD patients.

Glutamine, as one of the potent amino acids, can provide beneficial effects on protein synthesis of skeletal muscle [8]. Amino acid metabolism of skeletal muscles also generates glutamine to detoxify ammonia [9,10]. Moreover, it was previously demonstrated that glutamine maintains the integrity of the intestinal mucosal barrier and improves intestinal function [11,12]. In addition, glutamine is classified as a non-essential amino acid in healthy individuals [13], but abundant evidence suggests that glutamine is essential in specific stressful situations, such as severe illness, trauma, and overtraining [14,15,16,17,18,19]. However, research to reveal the effects of glutamine on ethanol-induced muscle loss is still limited.

Therefore, the hypothesis of this study was that glutamine supplementation may prevent dysbiosis and gut leakiness and thus may directly contribute to increased muscle protein synthesis and decreased muscle protein degradation in rats with chronic ethanol feeding. Furthermore, ethanol-induced liver injury may be ameliorated by glutamine supplementation, which also indirectly prevents disruption of muscle protein metabolism in rats. This animal study was performed to clarify the proposed hypotheses.

## 2. Materials and Methods

### 2.1. Animals

Male Wistar rats (BioLasco Taiwan, Ilan, Taiwan), aged 7 weeks, were used in this experiment. All rats were placed in individual stainless-steel cages. The animal room was maintained at 22 ± 2 °C with 50–70% humidity and a 12-h light–dark cycle. All procedures were approved by the Institutional Animal Care and Use Committee of Taipei Medical University (LAC-2017-0384). All rats were allowed free access to a standard rodent diet (LabDiet 5001 Rodent Diet; PMI Nutrition International, St. Louis, MO, USA) and water for 1 week of acclimation before the study.

### 2.2. Study Protocol

For the first 2 weeks, Wistar rats were divided into two groups and fed a control or glutamine-containing diet. For the following 6 weeks, rats fed the control diet were further divided into two groups according to whether the diet contained no ethanol (CC) or did contain ethanol (CE). Rats fed the glutamine-containing diet were also further divided into three groups, namely a GG group (glutamine-containing diet without ethanol), GE group (control diet with ethanol), and GEG group (glutamine-containing diet with ethanol). The diet composition in each group is shown in Table 1.

Urine and fecal sample were obtained before sacrifice. All rats were placed in a metabolic cage to collect urine samples 4–5 days before sacrifice, while feces were directly collected after defecation in week 8 of the experiment. At the end of the experiment, all rats were starved for 12 h and then anesthetized by an intraperitoneal injection of equal volumes of 1 mL/kg body weight (BW) Zoletil 50 (Virbac, Carros, France) and Rompun 20 (Bayer Korea, Seoul, Korea). Blood was drawn from the abdominal aorta, and the liver and intestinal tissues were removed. The gastrocnemius and quadriceps muscles of the rats were dissected out, and muscle weights on the left and right sides were measured. Blood samples were collected in heparin-containing tubes and centrifuged at 1200× *g* for 15 min at 4 °C to separate the plasma. All plasma and tissues were stored at −80 °C before being further analyzed.

### 2.3. Measurements and Analytical Procedures

#### 2.3.1. Assessment of Liver Damage

##### Liver Function Index

Aspartate aminotransferase (AST) and alanine aminotransferase (ALT) activities, as biomarkers of liver function, were assessed with the ADVIA^®^ 1800 Chemistry System (Siemens Healthcare Diagnostics, Eschborn, Germany).

##### Plasma Ammonia Concentration

The plasma concentration of ammonia was determined using a clinical biochemical analyzer with an enzymatic method of glutamate dehydrogenase (Dimension ExL, Siemens Healthcare, Erlangen, Germany).

##### Liver Histological Assessment

Hematoxylin and eosin (H&E) staining was used to observe pathological changes due to chronic liver injury, hepatocyte damage, steatosis, necrosis, and fibrosis. The H&E staining scale of the fatty liver score and inflamed liver score ranged from 0 to 4, with 0 representing the absence of, 1 trace, 2 mild, 3 moderate, and 4 severe fatty liver/inflammation [20].

##### Hepatic Cytokine Levels

Liver tissues were homogenized as described in a previous study [21]. The homogenized solution was centrifuged at 3000× *g* and 4 °C for 15 min. Commercial kits were used to analyze the supernatant, including the DuoSet^®^ rat tumor necrosis factor (TNF)-α kit, the rat interleukin (IL)-1β/IL-1F2 kit, the rat IL-6 kit, and the rat IL-10 kit (R&D Systems, Minneapolis, MN, USA). A microplate reader (Molecular Devices, Sunnyvale, CA, USA) was used to read the optical density (OD) at 450 nm for all cytokines.

##### Protein Expression of Cytochrome P450 2E1 (CYP2E1)

CYP2E1 quantification in liver tissue was performed using a Western blotting method based on procedures described by Uesugi et al. and Chen et al. [22,23].

##### Toll-Like Receptor 4 (TLR4) Signaling Pathway

The method of sample preparation of liver tissues was according to our previous study [21]. Sodium dodecylsulfate (SDS)-polyacrylamide gel electrophoresis (PAGE, 10%) was used to separate hepatic proteins (30 μg). Antibodies included a mouse monoclonal antibody to TRL-4 (IMG-5031A, Imgenex, San Diego, CA, USA) and a rabbit monoclonal antibody to myeloid differentiation primary response 88 (MyD88; D80F5, Cell Signaling Technology, Danvers, MA, USA).

#### 2.3.2. Assessment of Small-Intestinal Damage

##### Ratio of Urine Lactulose (L) to Mannitol (M)

In the last week of the experiment, the urine L/M ratio was determined for the intestinal permeability test [24]. Rats were fed using oral gavage with 0.5 mL of a lactulose (L) (100 mg/kg BW), mannitol (M) (6 mg/kg BW), and sucrose (200 mg/kg BW) solution and then starved for 8 h. Rats were placed in individual metabolic cages, and urine samples were collected after 6 h. Pooled urine samples were vortex-mixed for 1 min and centrifuged at 5000× *g* for 5 min to remove any sediment. Each urine sample was diluted with acetonitrile at a 1:4 ratio. Urine samples were analyzed with an AB SCIEX QTRAP^®^ high-performance liquid chromatography (HPLC) instrument (LC–MS/MS, AB SCIEX QTRAP^®^5500, Framingham, MA, USA). The L/M ratio in urine was calculated to determine the intestinal permeability.

##### Plasma Endotoxin Concentration

A Limulus Amebocyte Lysate Kit (Associates of Cape Cod, East Falmouth, MA, USA) was used to detect plasma endotoxin levels and check the OD at 450 nm.

##### Intestinal Bacterial Composition

The microbial distribution was analyzed using 16S ribosomal (r)RNA Next Generation Sequencing (NGS). Fecal DNA was extracted and purified using a magnetic bead-based nucleic acid purification method. Fresh feces from rats were collected into sterilized 2-mL Eppendorf tubes and stored at −80 °C for analysis. The V3 and V4 rRNA hypervariable gene regions were amplified two times with a PCR (polymerase chain reaction) Illumina MiSeq sequencing tool (Illumina, San Diego, CA, USA) with a length of 300 base pairs. A QIAamp^®^ Fast DNA Stool Mini Kit (Qiagen, Hilden, Germany) was used as the extraction kit. The DNA concentration and purity were determined. Extraction of 2 μL of sample DNA stock was performed with a wide-volume spectrum analyzer PowerWave XS2 (BioTek^®^, Winooski, VT, USA). The DNA gene pool for each sample was quantified by a Roche LightCycler 480 system quantitative (q)PCR. Overall genomics of the 16S rRNA database in the Greengenes database were used as a benchmark for classification. Original sequencing data were read, subjected to a quality control filter, and transformed. The QIIME package tool was used for bioinformatics microbiome analyses, including operational taxonomic unit (OTU) assignment, phylogenetic reconstruction, diversity analyses, and visualization. OTUs were assigned for each sequence based on 97% similarity.

#### 2.3.3. Assessment of Muscle Damage

##### Forelimb Grip Strength Test

Forelimb grip tests were determined using a grip strength tool (Model-RX-5, Aikoh Engineering, Nagoya, Japan). Tests were performed at weeks 2 and 8 of the experiment. Results were calculated as the average of three strength test assessments.

##### Muscle Histological Assessment

A histological assessment of paraffin sectioning was used for the gastrocnemius muscles. H&E staining was performed to differentiate collagen and connective tissues. Muscle damage was measured as a percentage of the entire cross-section area of the muscle as an absolute value in µm^2^.

##### Muscle Protein Synthesis Markers

The muscle protein synthesis marker, p70S6K1, was assessed by Western blotting methods. Quadricep muscle tissue of 0.1 g was homogenized in 1 mL RIPA buffer. The sample was centrifuged at 10^4^× *g* for 15 min at 4 °C. Muscle samples were mixed with 4× loading dye and boiled at 105 °C for 5 min. An amount of 60 µg of protein per sample was added to each well, and fresh running buffer (25 mM Tris base, 192 mM glycine, 20% (*v*/*v*) methanol, 0.037% SDS) was added to the gel electrophoresis tank (Mini PROTEAN^®^ Tetra Cell, Bio-Rad Laboratories Inc., Irvine, CA, USA). The primary antibody was anti-rabbit p70S6K (#9202, Cell Signaling Technology, Danvers, MA, USA). The secondary anti-rabbit immunoglobulin G (IgG) antibody (1:5000; C04003, Croyez Bioscience, Taipei, Taiwan) was added to Tris-buffered saline/Tween (TBST) at room temperature for 2 h. The membrane was washed three times with TBST (5 min/time). An H_2_O_2_: enhanced chemiluminescence (ECL) (1:1) solution (T-Pro LumiLong Plus Chemiluminescent Substrate Kit, T-Pro Biotechnology, New Taipei City, Taiwan) was added to the membrane, and an image was acquired using a UVP Chemidoc it 515 Imaging System (UVP LLC, Upland, CA, USA). The Western blot results were quantified using ImageJ Software and GAPDH as the internal control.

##### Muscle Protein Degradation Markers

As a protein degradation factor, myostatin further induces the Smad2/3 signaling mediator, which results in higher expression of MuRF1 [25]. Muscle protein degradation markers were analyzed by Western blot methods. The method of muscle sample preparation was the same as that for the measurement of muscle protein synthesis. The primary antibodies were the rabbit polyclonal myostatin (Protein Tech, Manchester, UK), rabbit monoclonal Smad2/3 (Cell Signaling Technology), and rabbit polyclonal MuRF1 (Cell Signaling Technology). An anti-rabbit IgG antibody (1:5000; C04003, Croyez Bioscience, Taipei, Taiwan) was used as the secondary antibody in this study.

#### 2.3.4. Amino Acid Composition

The method of extracting amino acids from plasma, liver, and muscle was modified from the protocol described by Yuan [26]. The amino acid analysis was performed using a liquid chromatographic/tandem mass spectroscopic (LC/MS/MS) system for plasma and ultra-performance liquid chromatography (UPLC)/MS system for liver and muscle tissues. The metabolite analysis was performed using an Acquity UPLC System (Waters, Milford, MA, USA) coupled with the Xevo TQ MS system (Waters). For UPLC, a 1.7-mm (2.13100 mm) C18 column (Acquity UPLC System, Waters, Santa Clara, CA, USA) was used. LC separation was carried out at 40 °C at a flow rate of 0.3 mL/min using the following gradient for the analysis: 0–0.5 min 1% B, 0.5–2.5 min from 1% B to 10% B, 2–3.5 min 10% to 35% B, 3.5–6 min from 35% to 99% B, and 6–9 min 1% B (solvent system A: water/formic acid (100:0.1, *v*/*v*); B: acetonitrile/formic acid (100:0.1, *v*/*v*). Data were acquired with TargetLynx software (Waters, Santa Clara, CA, USA).

#### 2.3.5. Statistical Analysis

Values are expressed as mean ± standard deviation (SD). Statistical analyses were performed with SPSS version 25 software (SPSS, Chicago, IL, USA). Differences between groups were compared by a one-way analysis of variance (ANOVA) and followed by Fisher’s least significant difference (LSD) post-hoc test. Statistical significance was assigned at the *p* < 0.05 level.

## 3. Results

### 3.1. Food Intake, Ethanol Consumption, Final BW, Relative Liver Weight, and Muscle Weights

Food intake showed no differences among the five groups (CC group: 81.4 ± 0.6 kcal/day; GG group: 81.6 ± 0.6 kcal/day; CE group: 79.8 ± 8.3 kcal/day; GE group: 78.6 ± 1.0 kcal/day; and GEG group: 76.9 ± 3.5 kcal/day). Average ethanol consumption levels in the CE, GE, and GEG groups were 3.7 ± 0.5, 3.6 ± 0.7, and 3.5 ± 0.2 g/day, respectively. There was no difference in the ethanol intake among these groups.

Final BWs and relative liver weights are shown in Table 2. Compared to the CC group, the final BW of the CE group was significantly lower (*p* < 0.05). However, there were no differences among the CE, GE, and GEG groups. The relative liver weight of the CE group was significantly higher than that of the CC group, but there were no differences among the ethanol-fed groups (Table 2). Furthermore, no change was observed in muscle weights, for either the quadriceps or gastrocnemius, among the groups (Table 2).

### 3.2. Assessment of Liver Damage

#### 3.2.1. Plasma AST and ALT Activities and Ammonia Level

As shown in Table 3, plasma AST and ALT activities in the CE group were significantly higher than those in the CC group (*p* < 0.05). However, plasma AST activities of the GE and GEG groups were significantly lower than those of the CE group (Table 3). The CE group presented the highest plasma ammonia level; however, plasma ammonia levels were significantly reduced in the GE and GEG groups.

#### 3.2.2. Hepatic Histopathology Scores and Cytokine Levels

Hepatic fatty changes and inflammatory scores were significantly elevated in the CE group, while those scores were significantly lower in the GE and GEG groups (Table 4). According to the hepatic histopathology, steatosis and inflammatory cell infiltration were observed in the CE group (Figure 1).

On the other hand, hepatic cytokine levels were significantly higher in the CE group than those in the CC group. However, hepatic cytokine levels were significantly decreased in the GE and GEG groups (Table 4).

#### 3.2.3. Hepatic CYP2E1

As shown in Figure 2, CYP2E1 protein expression was significantly increased in the CE group compared to the CC group. Moreover, there was no difference among ethanol-intake groups, i.e., CE, GE, and GEG groups.

#### 3.2.4. TLR4-Signaling Pathway

Results of the inflammatory signaling pathways of TLR4 and MyD88 showed that TLR4 and MyD88 protein expressions were significantly elevated in the CE group compared to the CC group; however, those of the GE group were significantly reduced compared to the CE group (Figure 3). There was no difference between the CE and GEG groups in TLR4 and MyD88 protein expressions (Figure 3).

### 3.3. Assessment of Gut Damage

#### 3.3.1. Intestinal Permeability

Intestinal permeability was evaluated using the L/M ratio in urine and the plasma endotoxin level (Table 5). A higher L/M ratio and plasma endotoxin level indicate gut leakiness. Results showed that the CE group had a significantly higher L/M ratio and endotoxin level compared to the CC group (Table 5). However, the L/M ratio and endotoxin level were lower in both the GE and GEG groups compared to the CE group (Table 5).

#### 3.3.2. Fecal Bacterial Composition

In order to determine intra-individual diversity, α-diversity parameters were analyzed in feces. It was found that there was no significant difference among all groups (Table 6).

According to the phylum classification level, no significant differences in Firmicutes or Bacteroidetes were found in all groups, and in terms of their ratios as illustrated in Figure 4. The Proteobacteria were reduced in the GG, GE, and GEG groups compared to the CC group. The GE group showed a lower Actinobacteria level than that of the GG group. The Tenericutes was higher in the CE group compared to the CC, GG, and GE groups.

To determine the degree of dissimilarity between groups, a β-diversity analysis was performed using a principal component analysis (PCA) plot as shown in Figure 5. Results showed that the microbiota distribution between groups did not distinctly intersect among groups.

To identify changes in specific bacterial taxonomical abundances between groups, a linear discriminant analysis effect size (LEfSe) approach and linear discriminant analysis (LDA) score were assessed in this study (Figure 6). It was found that the Erysipelotrichi class, Erysipelotrichales order, Erysipelotrichaceae family, *Streptococcus* genus and *Anaerostipes* genus of the Firmicutes phylum were commonly found in CC fecal samples. Other groups of bacteria were also abundant in the CC group including the *Desulfovibrio* genus and *Shigella sonnei* species from the Proteobacteria phylum. Moreover, the Rhizobiales order, Lactobacillus family, and *Klebsiella* genus were overrepresented in the GG group. At the species level, *Lactobacillus reuteri*, *Enterobacter arachidis*, and *Rothia nasimurium* species also had higher LDA scores in the GG group. In the CE group, the *Providencia* genus of the Proteobacteria phylum and Corynebacteriaceae class of the Actinobacteria phylum were commonly observed. In contrast, distinct microbes were also found in the GE and GEG groups. In the GE group, the Peptococaceae class of the Firmicutes phylum and *Bifidobacterium animalis* species of the Actinobacteria phylum were dominant bacteria. In contrast, the Bacillaceae family, Clostridia class, and *rc4_4* genus were overrepresented in the GEG group which belong to the Firmicutes phylum.

### 3.4. Assessment of Muscle Damage

#### 3.4.1. Forelimb Grip Strength Test

Muscle strength was evaluated using grip strength and showed no significant differences among all groups at the initial or final time points, or in grip strength adjusted by BW (Table 7).

#### 3.4.2. Muscle Histological Assessment

Histological features of muscles by H&E staining are illustrated in Figure 7. However, there were no differences in the muscle myofiber cross-sectional area (CSA) among the groups.

#### 3.4.3. Muscle Protein Synthesis Markers

The protein expression of the protein synthesis marker, p70S6K, was significantly reduced in the CE group compared to the CC group. Moreover, p70S6K protein expressions of the GE and GEE groups were significantly higher than those of the CE group (Figure 8A).

#### 3.4.4. Muscle Protein Degradation Markers

Differences in protein expressions of protein degradation factors, including myostatin, Smad2/3, and MuRF1, were not found among the groups (Figure 8B).

### 3.5. Amino Acid Composition

As shown in Appendix A, compared to the CC group, the plasma arginine level was lower in the CE group. However, no differences were found among the CE, GE, and GEG groups in plasma amino acid levels. In addition, the hepatic lysine level was higher, and valine, isoleucine, glutamine, glutamic acid, and tyrosine levels were lower in the CE group compared to the CC group (all *p* < 0.05). On the other hand, the GE and GEG groups presented significantly higher hepatic valine, isoleucine, and tyrosine levels compared to the CE group. Moreover, the hepatic glutamine level of the GEG group was significantly higher than that of the CC group. Although the muscle leucine level was significantly lower in the CE group than that in the CC group, it was significantly elevated in the GE and GEG groups.

## 4. Discussion

### 4.1. BW and Liver Weight

In the present study, rats fed ethanol showed significantly lower BWs and significantly heavier relative liver weights (Table 2). ALD patients are reported to have malnutrition problems, which cause muscle and BW losses [27]. Moreover, 90–95% of ALD patients have hepatomegaly [28]. Additionally, glutamine supplementation also showed no significant impact on the BWs of rats in either the ethanol (GG) or non-ethanol groups (GE, GEG), which is consistent with our previous study [29]. This result indicated that glutamine supplementation did not affect the BW or liver weight in rats undergoing chronic ethanol feeding.

### 4.2. Ethanol, Glutamine, and Liver Damage

After feeding the ethanol-containing diet for 6 weeks, fatty changes and inflammation accompanied by higher AST and ALT activities were observed (Table 3, Figure 1). Moreover, TLR4 and Myd88 protein expressions were stimulated, which caused higher cytokine levels (Figure 2 and Figure 3, Table 4). CYP2E1 protein expression was also elevated, which caused oxidative stress (Figure 2) [30]. Results of a liver damage assessment indicated that liver damage was successfully induced by chronic ethanol feeding in rats. This result was consistent with our previous studies [21,23,29]. On the other hand, glutamine supplementation ameliorated ethanol-induced liver injuries, such as inhibition of fatty changes and inflammation, and reductions in liver function indicators and hepatic cytokines (Figure 1 and Figure 2, Table 3 and Table 4). However, the preventive effects of liver damage were similar regardless of the time of glutamine treatment, even though significant downregulation of TLR4 and Myd88 protein expressions was only observed in the GE group (Figure 3). Two possibilities were considered for the inconsistency between the TLR4 pathway and hepatic cytokine levels that occurred in the GEG group. One is that lipopolysaccharide (LPS)-TLR4 recognition could attract other molecules, such as MyD88 and adaptor protein TIR-domain-containing adapter-inducing interferon-β (TRIF) [31]. The other is that TLR2 activation by chronic alcohol consumption also induced an inflammatory pathway and was mediated by MyD88, which could produce proinflammatory cytokines [32]. Thus, protein expressions of the TRIR and TLR2 pathways should be measured in future studies. On the other hand, it is well known that glutamine is classified as a conditionally essential amino acid and is a beneficial factor for maintaining the integrity of gut membranes and preventing alterations of the gut microbiota [33]. It was speculated that the protective effects of glutamine on ethanol-induced liver damage might be due to maintaining the gut health in rats. Therefore, relationships among ethanol intake, glutamine, and gut damage are discussed below.

The plasma and hepatic TG levels were also measured (Appendix A). However, no difference was found in plasma TG level among groups. In addition, the hepatic TG level only showed an increasing trend in the CE group when compared to the CC group. In our previous study, it was found that the hepatic TG level was significantly increased in rats fed with ethanol-containing diet for 8 weeks [21,34]. In this study, the ethanol feeding period was only 6 weeks. The shorter feeding period might be the reason why plasma and hepatic TG levels did not change. However, the significant differences in fat accumulation were observed among groups based on the hepatic histological examination (Figure 1). The hepatic histological examination was organized as the direct evidence of liver damage; therefore, we used the hematoxylin A and eosin (H&E) staining of liver tissue sections and fatty change score as the indicators of fat accumulation in this study.

### 4.3. Ethanol, Glutamine, and Gut Damage

Intestinal damage was examined in this study with several indicators, such as intestinal permeability and the intestinal microbiota composition. Intestinal permeability was assessed using the urine L/M ratio. This ratio was highly correlated with intestinal permeability, as a higher excretion rate of the monosaccharide mannitol compared to the disaccharide lactulose, which is harder to be absorbed by the intestinal lumen. In this study, it was found that intestinal permeability increased, which caused higher blood endotoxin levels in rats fed the ethanol-containing diets (Table 5, CC vs. CE groups). In previous studies, chronic alcohol intake increased the intestinal permeability and decreased tight junction expressions [35,36,37,38]. Endotoxins, which are gut-derived bacterial products, can activate hepatic Kupffer’s cells in rats chronically fed ethanol, such that these cells overproduce cytokines such as TNF-α, IL-1β, IL-6, and IL-8 [39]. These cytokines not only directly injure hepatocytes, but they also initiate a hepatic necro-inflammatory cascade in the liver [40]. On the other hand, it was found that there was little to no change after 6 weeks of daily alcohol feeding in the ratio of Firmicutes and Bacteroidetes (F/B ratio), richness, or α- and β-diversities in this study (Table 6, Figure 4 and Figure 5). In our previous study, the F/B ratio increased after 8 weeks of ethanol feeding in rats [21]. Thus, when our prior studies are considered in the context of this study, the findings render support to a hypothesis that gut leakiness occurred earlier than dysbiosis in rats undergoing chronic ethanol feeding; in addition, endotoxemia could be caused by both gut leakiness and a dysbiotic microbiome in alcoholics. As to the abundances of specific bacteria, it was observed that greater abundances in the family Corynebacteriaceae and the genus *Providencia bacteria* were found in ethanol-fed rats (Figure 5). Increasing *Corynebacterium* bacteria may indicate ALD in mouse studies [35]. In the past, multiple studies indicated that opportunistic *Corynebacterium* infections also occurred in alcoholic liver patients [41,42].

When rats were supplemented with glutamine, the L/M ratio was reduced, which occurred with lower blood endotoxin levels in rats fed ethanol (Table 5, CC vs. GE and GEG groups). This indicated that glutamine supplementation inhibited gut leakage in rats that ingested ethanol. Past studies revealed that glutamine improved the intestinal permeability and maintained the intestinal integrity function [43]. Glutamine prevented acetaldehyde-induced disruption of tight and adherens junctions in human colon cells [44] and in Caco-2 cells [45]. In an animal model, glutamine attenuated intestinal mucosal injury and promoted mucosal recovery in lipoprotein saccharide (LPS)-treated rats [46]. Glutamine increased expressions of tight junction and adherens junction proteins in mice [47] and rats [48] chronically fed alcohol. Moreover, the lower blood endotoxin level caused by glutamine treatment was connected to the reduction of inflammation in ethanol-induced liver injury based on the viewpoint of the gut–liver axis. On the other hand, glutamine supplementation did not change the α-diversity or β-diversity in ethanol-fed rats, but some dominant bacteria were found in the GE and GEG groups (Figure 5). Certain *Bifidobacteria* bacteria (*Bifidobacteria animalis*) were higher in the GE group, which was linked to preservation of the gut barrier function and the production of butyrate [49,50]. *Clostridia* bacteria significantly increased in the GEG group, which is also beneficial as a producer of butyrate [50]. Butyrate, an essential metabolite in the colon, can maintain the gut barrier function and has immunomodulatory and anti-inflammatory properties [51].

### 4.4. Ethanol, Glutamine, and Muscular Protein Synthesis and Degradation

Several variables in muscle loss were analyzed in this study, including muscle strength, myofiber size, and also protein synthesis and degradation pathways. No significant differences in muscle weight, grip strength, or myofiber size were found in rats after feeding ethanol-containing diets for 6 weeks (Table 2 and Table 7, Figure 7, CC vs. CE group). Previous studies showed that excessive alcohol consumption can decrease the muscle mass and CSA specifically of type II fiber-rich muscles, which are categorized as fast fibers [52,53,54,55,56]. The lean body mass and gastrocnemius muscle mass were profoundly reduced after sustained alcohol consumption in aged female rats [57].

However, it was found that p70S6K protein expression as a protein synthesis marker in muscles was reduced in rats fed ethanol (Figure 8A, CC vs. CE group). Past studies also indicated that protein synthesis decreased along with protein synthesis markers, including phosphorylated (p)-4EBP1, p-eIF4G, p-S6, and p-p70S6K [37]. It was speculated that the ethanol-feeding period in this study was too short to induce muscle loss in rats, although the protein synthesis marker decreased. On the other hand, the protein synthesis process in muscles is mainly initiated by insulin-like growth factor 1 (IGF1). Insulin or IGF1 can directly bind to the IGF receptor and phosphorylate insulin receptor substrate (IRS)-1 that acts as an intracellular receptor [58]. IRS-1 phosphorylation impacts phosphatidylinositol 3-kinase (PI3K) translocation and phosphorylation [56]. Moreover, activation of PI3K can stimulate activation of serine/threonine kinase protein kinase B (Akt), which further stimulates its downstream signaling molecules, such as mammalian target of rapamycin (mTOR), which can upregulate protein synthesis signaling molecules, such as p70S6K1 [59,60]. Therefore, the PI3K-Akt-mTOR signaling pathway for muscular protein synthesis should be measured in future studies.

In this study, protein expressions of muscle protein degradation markers, such as myostatin, MuRF1, and Smad2/3, did not change in rats under chronic ethanol intake (Figure 8B, CC vs. CE group). This might be another reason why muscle loss was not observed in rats fed ethanol for 6 weeks in this study. Myostatin induces the Smad2/3 signaling mediator, which results in higher protein expressions of MuRF1 and atrogin-1 [61]. Past studies showed that both atrogin-1 and MuRF1 messenger (m) RNA expressions were increased in muscles after alcohol consumption in adult and aged animals [56]. Cohen et al. indicated that mTOR activation downregulated the ubiquitin proteasome system (UPS), which might be related to protein breakdown in muscles [62]. Thus, the mTOR pathway and UPS of skeletal muscles are important issues in ethanol-fed rats.

It was found that glutamine supplementation did not increase muscle weight, strength, or muscle size in alcohol-fed rats (Table 2 and Table 7, Figure 7, CE vs. GE and GEG groups). However, p70S6K1 protein expression was significantly higher in rats chronically fed ethanol and supplemented with glutamine (Figure 8A, CE vs. GE and GEG groups). In addition, the muscle protein degradation markers did not change in rats chronically fed ethanol with glutamine supplementation (Figure 8B, CE vs. GE and GEG groups). The current study results are consistent with previous studies that found p70S6k protein expression increased after glutamine supplementation [19,63]. It was previously reported that the p70S6K level significantly increased by the administration of BCAAs in combination with glutamine supplementation but did not change when supplemented with glutamine or BCAAs alone in rats with a total gastrectomy [63]. Glutamine also stimulated protein synthesis in human primary myotubes via an increment in the p-p70S6K level [19].

### 4.5. Glutamine and Gut–Muscle Axis with Chronic Ethanol Intake

There are two key results connecting muscle damage with gut health in this study. First is the diminished effect of glutamine on the blood endotoxin level induced by ethanol intake. It was previously demonstrated that endotoxin caused skeletal muscle atrophy in sepsis syndrome patients and ALD patients [5,6,64]. Therefore, it is thought that the administration of glutamine promotes protein expression by alleviating high blood endotoxin levels because of improved gut permeability. Second, glutamine increased the abundances of Bifidobacteria and Clostridia, which were reduced by chronic ethanol feeding, thus enhancing the production of short chain fatty acids (SCFAs), especially butyrate. Frampton et al. indicated that SCFAs are absorbed from the gut lumen and modulate host metabolic responses at different organ sites, including skeletal muscles, the largest organ in humans [65]. Therefore, the possible reason that glutamine improved the muscle protein synthesis factor in rats under ethanol feeding was closely related to the production of SCFAs in the gut. The measurement of fecal SCFA levels is needed in future studies.

### 4.6. Glutamine and the Liver–Muscle Axis with Chronic Ethanol Intake

This study demonstrated that higher blood ammonia was found in rats fed ethanol for 6 weeks (Table 3, CC vs. CE group). It was also found that the low hepatic glutamine levels also decreased in rats fed ethanol (Supplement, CC vs. CE group). Hyperammonemia is one of the indications of altered ammonia metabolism and commonly occurs in alcoholic liver patients [66]. Elevated plasma ammonia levels are caused by decreased liver function and lead to disruption of urea cycle synthesis [66]. It was assumed that this could lead to a vicious cycle of glutamine and ammonia in cirrhotic patients [67]. On the other hand, hepatic valine and isoleucine levels were decreased in the CE group; in addition, the muscle leucine level was lower as well (Supplement, CC vs. CE group). It was indicated that BCAAs (valine, leucine, and isoleucine) for ammonia detoxification to glutamine in muscles is the cause of decreased BCAA levels in liver cirrhosis and urea disorders [68]. Moreover, it was demonstrated that BCAAs act as a significant energy substrate for muscles in several hypermetabolic states, such as sepsis, burn injuries, trauma, and cancer [68]. Taken together, it was speculated that chronic ethanol intake might induce BACC catabolism in muscles, which is linked to hypermetabolic stress, such as ammonia detoxification.

However, hyperammonemia was inhibited when ethanol-fed rats were supplemented with glutamine (Table 3, CE vs. GE and GEG groups). Additionally, hepatic valine and isoleucine and muscle leucine levels were also increased in rats fed a glutamine-containing ethanol diet (Appendix A, CE vs. GE and GEG groups). As mentioned above, glutamine plays an important role in the process of ammonia detoxification as a nitrogen shuttle [69]. Glutamine can be synthesized from ammonia and glutamate with the help of glutamine synthase as a catalyst [67]. Therefore, it is reasonable to assume that the oral administration of supplemental glutamine resulted in sufficient plasma and intramuscular concentrations of amino acids to alter the activities and/or expressions of signaling molecules in the direction of increasing protein synthesis and decreasing proteolysis of muscles.

### 4.7. The Study Limitation

This study still has several limitations. Some biomarkers should be measured to clarify the relationship between glutamine supplementation and muscular protein metabolism (synthesis and degradation) in future studies, such as the PI3K-Akt-mTOR signaling pathway, the mTOR pathway, and the UPS of skeletal muscles. Regarding the regulatory effects of glutamine on gut microbiota composition, the fecal SCFA levels must be analyzed in future studies. Moreover, this study only discussed the pre-administration effects of glutamine on chronic ethanol feeding in rats. In future studies, it is necessary to discuss the therapeutic effects of glutamine for rats that already display ethanol-induced damage. Additionally, following the study of Yeh et al. [70], part of the casein was replaced by glutamine, which provided 25% nitrogen of the total amino acids in glutamine-containing liquid diet (0.84% glutamine). The different dosages of glutamine supplementation should be confirmed in future studies.

## 5. Conclusions

Results of this study indicated that glutamine improved the protein synthesis factor, p70S6K, in muscle tissues when rats were fed an ethanol-containing diet. The improvement in muscular protein synthesis by glutamine may have been due to three major pathways (Figure 9). First, when rats chronically ingested ethanol, glutamine reduced blood endotoxin levels by maintaining the intestinal permeability and a well-balanced microbiota composition, which inhibited muscle damage. Second, glutamine inhibited the elevation of blood ammonia levels because ethanol-induced liver injury was ameliorated. Last, glutamine regulated amino acid metabolism in both the liver and muscles.

## Figures and Tables

**Figure 1 nutrients-13-02788-f001:**
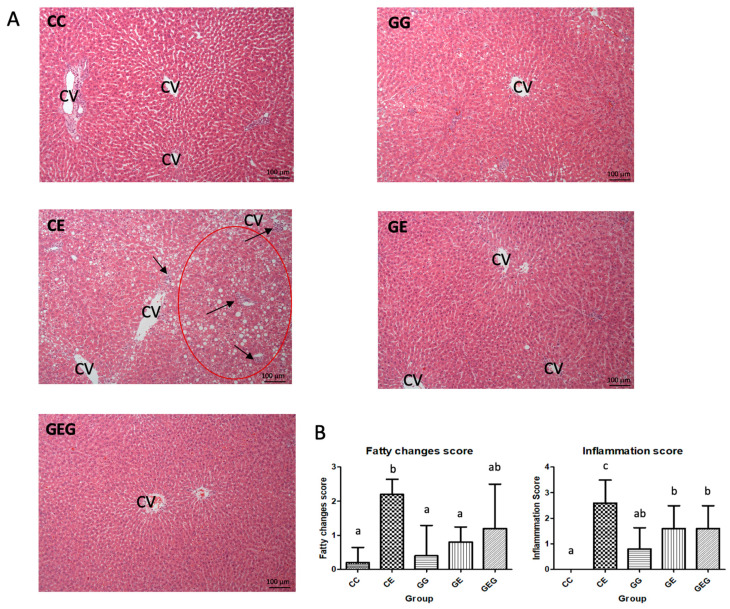
Effects of glutamine on hematoxylin A and eosin (H&E) staining of liver tissue sections in rats with chronic ethanol feeding. CV, central vein. (**A**) H&E staining showed fatty changes (red circle), hepatocyte degeneration, and necrosis accompanied by inflammatory cell infiltration (black arrow) in the CE group. (**B**) The pathological scores of fatty change and inflammation. Values are expressed as the mean ± SD. Means with different superscript letters in the same column significantly differ (*p* < 0.05). The CC group was fed the control diet for 8 weeks; the GG group was fed a glutamine-containing diet for 8 weeks; the CE group was fed the control diet the first 2 weeks and then an ethanol-containing diet for the next 6 weeks; the GE group was fed a glutamine-containing diet the first 2 weeks and then an ethanol-containing control diet for the next 6 weeks; the GEG group was fed a glutamine-containing diet for the first 2 weeks, and then a glutamine-containing diet with ethanol for the next 6 weeks.

**Figure 2 nutrients-13-02788-f002:**
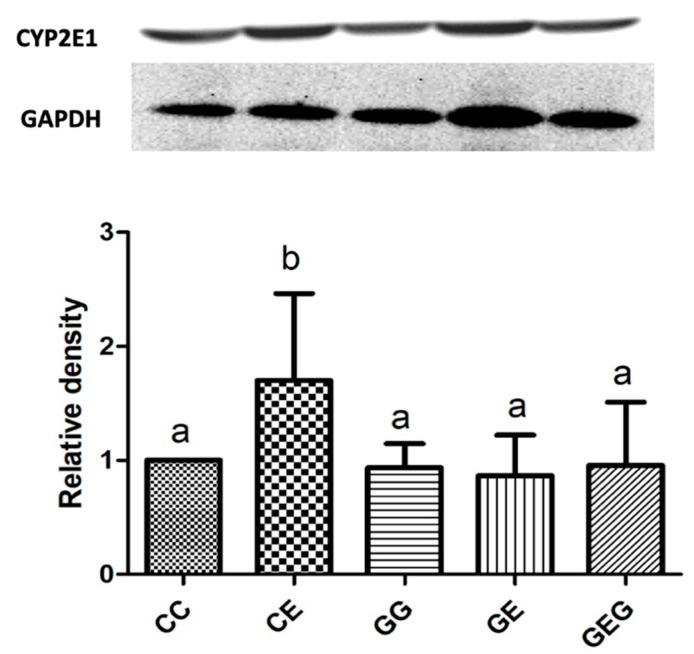
Effects of glutamine supplementation on CYP2E1 protein expression in chronic ethanol-fed rats. Quantification of Western blot on CYP2E1 level as density ratio of target protein compared to GAPDH (internal control) was calculated by setting the value of CC group as 1. CYP2E1, cytochrome P450 2E1; GAPDH, glyceraldehyde 3-phosphate dehydrogenase. The CC group was fed the control diet for 8 weeks; the GG group was fed a glutamine-containing diet for 8 weeks; the CE group was fed the control diet the first 2 weeks and then an ethanol-containing diet for the next 6 weeks; the GE group was fed a glutamine-containing diet the first 2 weeks and then an ethanol-containing control diet for the next 6 weeks; the GEG group was fed a glutamine-containing diet for the first 2 weeks, and then a glutamine-containing diet with ethanol for the next 6 weeks. The mean with different superscript letters in the same column was significantly different (*p* < 0.05).

**Figure 3 nutrients-13-02788-f003:**
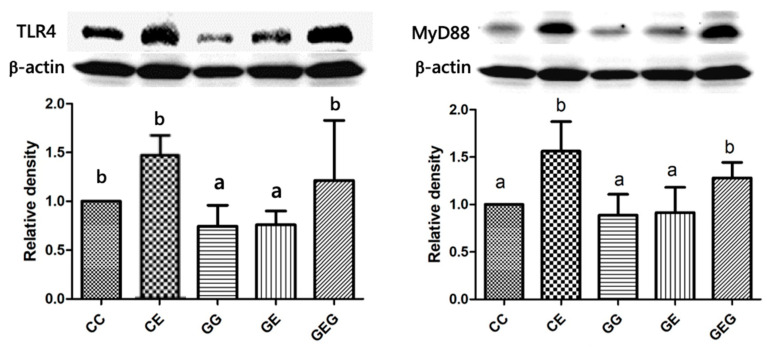
Effects of glutamine supplementation on the protein expressions of TLR4 and MyD88 in chronic ethanol-fed rats. Quantification of Western blot on TLR4 level as density ratio of target protein compared to β-actin (internal control) was calculated by setting the value of CC group as 1. TLR4, toll-like receptor 4; MyD88, myeloid differentiation primary response 88. The CC group was fed the control diet for 8 weeks; the GG group was fed a glutamine-containing diet for 8 weeks; the CE group was fed the control diet the first 2 weeks and then an ethanol-containing diet for the next 6 weeks; the GE group was fed a glutamine-containing diet the first 2 weeks and then an ethanol-containing control diet for the next 6 weeks; the GEG group was fed a glutamine-containing diet for the first 2 weeks, and then a glutamine-containing diet with ethanol for the next 6 weeks. The mean with different superscript letters in the same column was significantly different (*p* < 0.05).

**Figure 4 nutrients-13-02788-f004:**
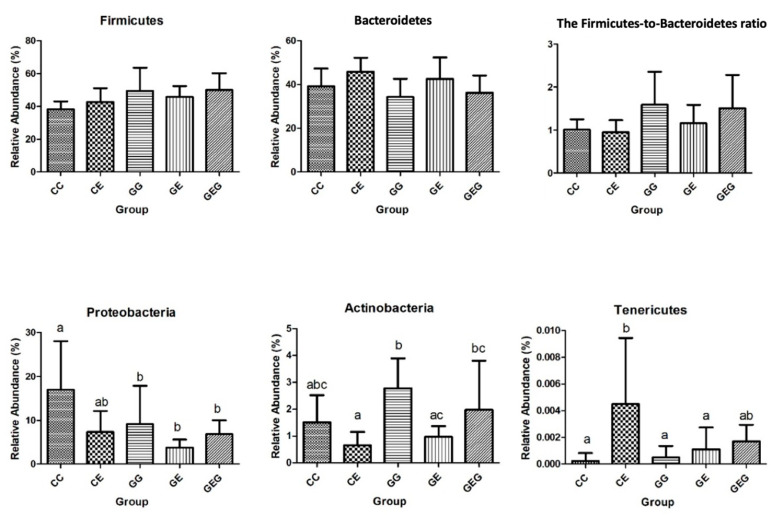
Effects of glutamine on the relative abundance of phyla in chronic ethanol-fed rats. The CC group was fed the control diet for 8 weeks; the GG group was fed a glutamine-containing diet for 8 weeks; the CE group was fed the control diet the first 2 weeks and then an ethanol-containing diet for the next 6 weeks; the GE group was fed a glutamine-containing diet the first 2 weeks and then an ethanol-containing control diet for the next 6 weeks; the GEG group was fed a glutamine-containing diet for the first 2 weeks, and then a glutamine-containing diet with ethanol for the next 6 weeks.

**Figure 5 nutrients-13-02788-f005:**
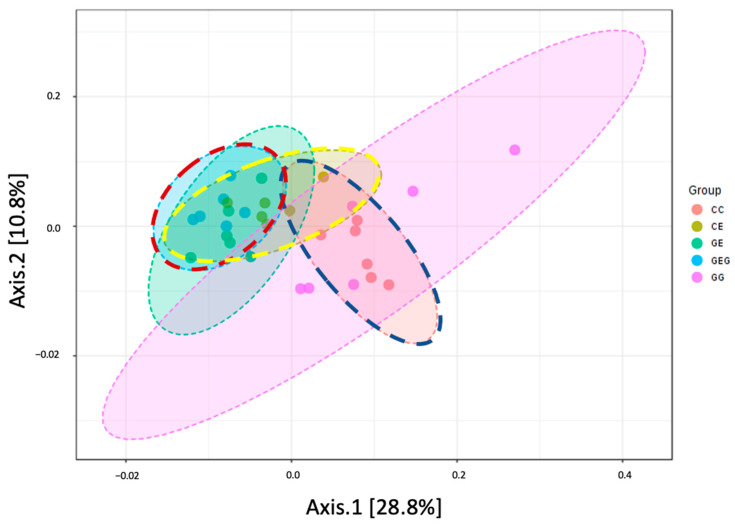
Effects of glutamine on principal coordinate analysis (PCoA) plot of bacterial distribution in chronic ethanol-fed rats. The CC group was fed the control diet for 8 weeks; the GG group was fed a glutamine-containing diet for 8 weeks; the CE group was fed the control diet the first 2 weeks and then an ethanol-containing diet for the next 6 weeks; the GE group was fed a glutamine-containing diet the first 2 weeks and then an ethanol-containing control diet for the next 6 weeks; the GEG group was fed a glutamine-containing diet for the first 2 weeks, and then a glutamine-containing diet with ethanol for the next 6 weeks.

**Figure 6 nutrients-13-02788-f006:**
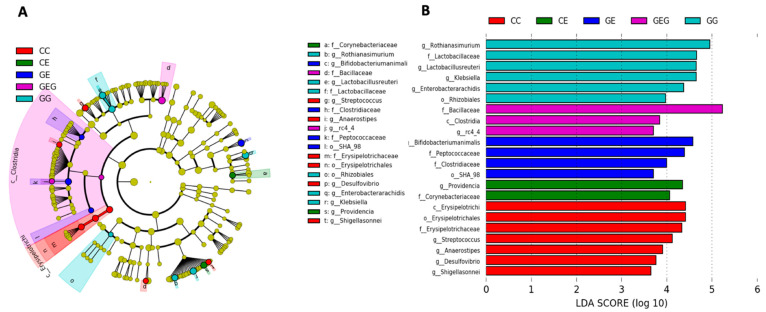
Effects of glutamine on specific bacterial taxonomical abundant changes in chronic ethanol-fed rats. (**A**) Linear discriminant analysis effect size (LeFSe) cladogram analysis. The center inner circle to outer circle part represented taxonomic levels from phylum to genus. (**B**) Discriminative biomarkers in various taxonomical level between group in linear discriminant analysis (LDA) scores in log10 scale. The CC group (red) was fed the control diet for 8 weeks; the GG group (aqua) was fed a glutamine-containing diet for 8 weeks; the CE group (green) was fed a control diet in the first two weeks and then an ethanol-containing diet for next six weeks; the GE group (blue) was fed a glutamine-containing diet in the first two weeks and ethanol-containing diet for next six weeks; the GEG group (purple) was fed a glutamine-containing diet for first two weeks and then a glutamine-containing diet with ethanol for the next six weeks.

**Figure 7 nutrients-13-02788-f007:**
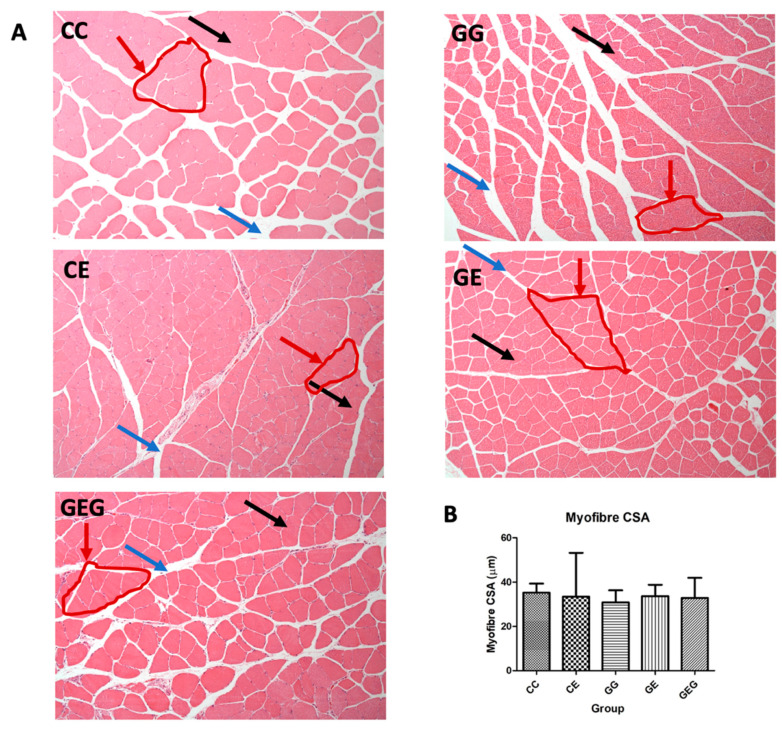
Effects of glutamine supplementation on histological muscle myofiber cross sectional area in chronic ethanol-fed rats. CSA, cross sectional area. (**A**) Histological representative of muscle tissue with hematoxylin and eosin (H&E) staining. Black arrows indicated the muscle fiber, blue arrows indicated the perimysium, and red arrows represented muscle fascicle. (**B**) The quantitative analysis of muscle myofiber cross sectional area. The CC group was fed the control diet for 8 weeks; the GG group was fed a glutamine-containing diet for 8 weeks; the CE group was fed the control diet the first 2 weeks and then an ethanol-containing diet for the next 6 weeks; the GE group was fed a glutamine-containing diet the first 2 weeks and then an ethanol-containing control diet for the next 6 weeks; the GEG group was fed a glutamine-containing diet for the first 2 weeks and then a glutamine-containing diet with ethanol for the next 6 weeks.

**Figure 8 nutrients-13-02788-f008:**
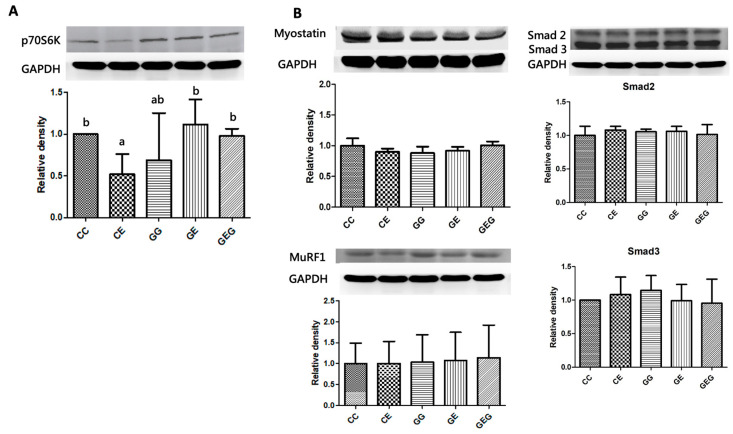
Effects of glutamine supplementation on muscular protein synthesis and degradation in chronic ethanol-fed rats. (**A**) As the protein synthesis factor, quantification of Western blot on the p70S6K level as the density ratio of the target protein compared to GAPDH (internal control) was calculated by setting the value of the CC group as 1. (**B**) The protein degradation factors were analyzed, including myostatin, Smad2/3, and MuRF1. Quantification of Western blot on myostatin, Smad2/3, and MuRF1 as the density ratio of the target protein compared to GAPDH (internal control) was calculated by setting the value of CC group as 1. p70S6K: ribosomal protein S6 kinase; MuRF1, muscle RING-finger protein-1; GAPDH, glyceraldehyde 3-phosphate dehydrogenase. The CC group was fed the control diet for 8 weeks; the GG group was fed a glutamine-containing diet for 8 weeks; the CE group was fed the control diet the first 2 weeks and then an ethanol-containing diet for the next 6 weeks; the GE group was fed a glutamine-containing diet the first 2 weeks and then an ethanol-containing control diet for the next 6 weeks; the GEG group was fed a glutamine-containing diet for the first 2 weeks and then a glutamine-containing diet with ethanol for the next 6 weeks.

**Figure 9 nutrients-13-02788-f009:**
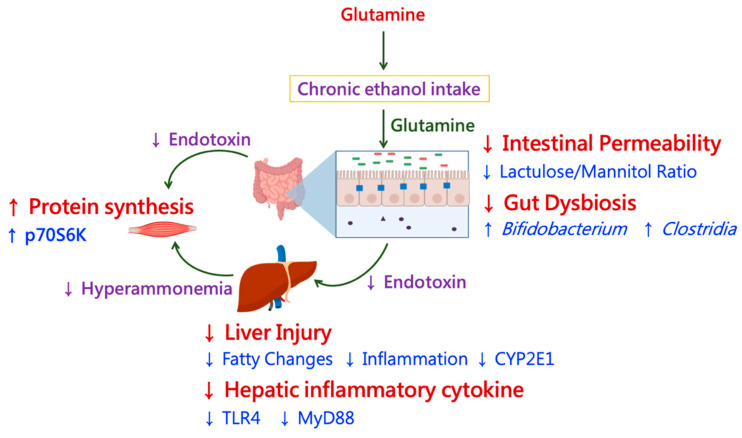
The prophylactic effects of glutamine supplementation on improving muscular protein synthesis in chronic ethanol-fed rats. (1) In this study, it was indicated that glutamine improved the protein synthesis in muscle tissue when rats were fed with an ethanol-containing diet. (2) The improvement of muscular protein synthesis by glutamine may be due to three major pathways. First, when rats chronically ingested ethanol, glutamine reduced blood endotoxin level by maintaining the intestinal permeability and well-balanced microbiota composition, which inhibited muscle damage. Secondly, glutamine inhibited the elevation of blood ammonia level because ethanol-induced liver injury was ameliorated. Lastly, glutamine regulated the amino acid metabolism both in the liver and muscle.

**Table 1 nutrients-13-02788-t001:** Experimental diet composition ^1^.

Ingredient ^2^	Diet Type			
	Control	Glutamine ^3^	Ethanol	Ethanol + Glutamine ^3^
	(g/L (1000 kcal))			
Casein	41.4	31.03	41.4	31.03
Maltodextrin	115.2	117.1	25.6	27.5
ICN: AIN-76 vitamins	2.5	2.5	2.5	2.5
ICN: AIN-76 minerals	2.6	2.6	2.6	2.6
L-Cysteine	0.5	0.5	0.5	0.5
DL-Methionine	0.3	0.3	0.3	0.3
Olive oil	28.4	28.4	28.4	28.4
Corn oil	8.5	8.5	8.5	8.5
Safflower oil	2.7	2.7	2.7	2.7
Choline bitartrate	0.53	0.53	0.53	0.53
Fiber	10	10	10	10
Xanthan gum	3	3	3	3
Ethanol	-	-	50	50
Glutamine	-	8.437	-	8.437
Total nitrogen	6.47	6.47	6.47	6.47

^1^ The CC group received the control diet for 8 weeks; the GG group received the glutamine-containing diet for 8 weeks; the CE group received the control diet for the first 2 weeks and an ethanol-containing diet for the next 6 weeks; the GE group received the glutamine-containing diet for the first 2 weeks and an ethanol-containing control diet for the next 6 weeks; the GEG group received the glutamine-containing diet for the first 2 weeks, and then the glutamine-containing diet with ethanol for the next 6 weeks. ^2^ Casein, AIN-76 vitamins, AIN-76 minerals, L-cysteine, DL-methionine, choline bitartrate, fiber, and maltodextrin were purchased from ICN Biochemicals (Costa Mesa, CA, USA). Xanthan gum, ethanol, and glutamine were acquired from Sigma-Aldrich (St. Louis, MO, USA). Corn oil and olive oil were obtained from God Bene Enterprise (Yunlin, Taiwan). Safflower oil was purchased from Taiwan Sugar Corporation (Taipei, Taiwan). ^3^ Both the glutamine-containing diet and glutamine-containing diet with ethanol contained 0.84% glutamine.

**Table 2 nutrients-13-02788-t002:** Effects of glutamine on final body weight, liver weight, and muscle weight in rats chronically fed ethanol ^1,2^.

Group	Final Body Weight (g)	Relative Liver Weight ^3^ (%)	Quadriceps Weight (g)	Gastrocnemius Weight (g)
CC	422.9 ± 7.5 ^a^	2.4 ± 0.2 ^a^	2.5 ± 0.4	3.5 ± 0.4
CE	389.1 ± 34.9 ^bc^	2.9 ± 0.2 ^b^	2.7 ± 0.5	3.2 ± 0.5
GG	407.1 ± 16.1 ^ab^	2.3 ± 0.2 ^a^	2.4 ± 0.6	3.3 ± 0.6
GE	375.8 ± 23.7 ^c^	2.8 ± 0.2 ^b^	2.2 ± 0.4	3.3 ± 0.4
GEG	381.3 ± 19.2 ^c^	2.7 ± 0.2 ^b^	2.3 ± 0.5	3.0 ± 0.3

^1^ Values are expressed as the mean ± SD. Means with different superscript letters in the same column significantly differ (*p* < 0.05). ^2^ The CC group was fed the control diet for 8 weeks; the GG group was fed a glutamine-containing diet for 8 weeks; the CE group was fed the control diet the first 2 weeks and then an ethanol-containing diet for the next 6 weeks; the GE group was fed a glutamine-containing diet the first 2 weeks and then an ethanol-containing control diet for the next 6 weeks; the GEG group was fed a glutamine-containing diet for the first 2 weeks, and then a glutamine-containing diet with ethanol for the next 6 weeks. ^3^ Relative liver weight: (liver weight/body weight) × 100%. CC: contained no ethanol; CE: contain ethanol; GG: glutamine-containing diet without ethanol; GE: control diet with ethanol; GEG: glutamine-containing diet with ethanol.

**Table 3 nutrients-13-02788-t003:** Effects of glutamine on plasma aspartate aminotransferase (AST) and alanine aminotransferase (ALT) activities and ammonia levels in rats chronically fed ethanol ^1,2^.

Group	AST (U/L)	ALT (U/L)	Ammonia (μg/dL)
CC	73.9 ± 7.4 ^a^	43.3 ± 6.8 ^a^	175.1 ± 22.1 ^a^
CE	130.4 ± 33.5 ^c^	113.4 ± 65.5 ^b^	221.7 ± 85.0 ^b^
GG	74.3 ± 4.8 ^a^	39.6 ± 7.1 ^a^	152.1 ± 22.0 ^ac^
GE	94.8 ± 22.5 ^b^	65.6 ± 19.3 ^a^	99.4 ± 33.3 ^cd^
GEG	100.6 ± 12.6 ^b^	54.1 ± 14.7 ^a^	70.3 ± 19.8 ^d^

^1^ Values are expressed as the mean ± SD. Means with different superscript letters in the same column significantly differ (*p* < 0.05). ^2^ The CC group was fed the control diet for 8 weeks; the GG group was fed a glutamine-containing diet for 8 weeks; the CE group was fed the control diet the first 2 weeks and then an ethanol-containing diet for the next 6 weeks; the GE group was fed a glutamine-containing diet the first 2 weeks and then an ethanol-containing control diet for the next 6 weeks; the GEG group was fed a glutamine-containing diet for the first 2 weeks, and then a glutamine-containing diet with ethanol for the next 6 weeks.

**Table 4 nutrients-13-02788-t004:** Effects of glutamine on hepatic inflammatory cytokines in chronic ethanol-fed rats ^1,2^.

Groups	IL-1β (pg/mg Protein)	IL-6 (pg/mg Protein)	IL-10 (pg/mg Protein)	TNF-α (pg/mg Protein)
CC	29.0 ± 4.9 ^a^	111.6 ± 26.0 ^a^	48.3 ± 6.6 ^a^	19.7 ± 2.0 ^a^
CE	53.1 ± 3.6 ^b^	149.3 ± 18.1 ^b^	72.6 ± 15.7 ^b^	32.8 ± 4.2 ^b^
GG	28.7 ± 3.1 ^a^	114.6 ± 24.7 ^a^	49.6 ± 13.2 ^a^	20.8 ± 3.6 ^a^
GE	28.9 ± 6.1 ^a^	106.7 ± 33.2 ^a^	50.8 ± 15.9 ^a^	16.9 ± 4.6 ^a^
GEG	28.2 ± 6.5 ^a^	104.2 ± 31.9 ^c^	60.4 ± 12.0 ^ac^	18.4 ± 3.8 ^a^

^1^ Values were expressed as the mean ± SD. The mean with different superscript letters in the same column was significantly different (*p* < 0.05). ^2^ The CC group was fed the control diet for 8 weeks; the GG group was fed a glutamine-containing diet for 8 weeks; the CE group was fed the control diet the first 2 weeks and then an ethanol-containing diet for the next 6 weeks; the GE group was fed a glutamine-containing diet the first 2 weeks and then an ethanol-containing control diet for the next 6 weeks; the GEG group was fed a glutamine-containing diet for the first 2 weeks, and then a glutamine-containing diet with ethanol for the next 6 weeks.

**Table 5 nutrients-13-02788-t005:** Effects of glutamine on intestinal permeability in chronic ethanol-fed rats ^1,2^.

Group	Ratio of Lactulose to Mannitol	Endotoxin (EU/mL)
CC	0.9 ± 0.0 ^a^	22.7 ± 3.3 ^a^
CE	3.0 ± 0.0 ^c^	29.7 ± 5.3 ^b^
GG	2.1 ± 0.0 ^b^	24.1 ± 4.5 ^a^
GE	2.6 ± 0.0 ^d^	23.4 ± 4.2 ^a^
GEG	1.0 ± 0.0 ^e^	20.7 ± 3.3 ^a^

^1^ Values were expressed as the mean ± SD. The mean with different superscript letters in the same column was significantly different (*p* < 0.05). ^2^ The CC group was fed the control diet for 8 weeks; the GG group was fed a glutamine-containing diet for 8 weeks; the CE group was fed the control diet the first 2 weeks and then an ethanol-containing diet for the next 6 weeks; the GE group was fed a glutamine-containing diet the first 2 weeks and then an ethanol-containing control diet for the next 6 weeks; the GEG group was fed a glutamine-containing diet for the first 2 weeks, and then a glutamine-containing diet with ethanol for the next 6 weeks.

**Table 6 nutrients-13-02788-t006:** Effects of glutamine supplementation on alpha diversity parameters in chronic ethanol-fed rats ^1,2^.

Group	Shannon	Simpson	ACE	Chao1
CC	3.1 ± 0.2	0.9 ± 0.0	518.4 ± 24.9	520.3 ± 21.7
CE	3.2 ± 0.3	0.8 ± 0.0	545.2 ± 17.7	553.7 ± 23.7
GG	3.3 ± 0.3	0.9 ± 0.0	514.2 ± 61.4	529.9 ± 67.7
GE	3.3 ± 0.5	0.8 ± 0.1	565.5 ± 14.2	568.4 ± 15.5
GEG	3.5 ± 0.2	0.9 ± 0.0	553.0 ± 24.8	561.9 ± 21.6

^1^ Values were expressed as the mean ± SD. ACE: abundance-based coverage estimator. ^2^ The CC group was fed the control diet for 8 weeks; the GG group was fed a glutamine-containing diet for 8 weeks; the CE group was fed the control diet the first 2 weeks and then an ethanol-containing diet for the next 6 weeks; the GE group was fed a glutamine-containing diet the first 2 weeks and then an ethanol-containing control diet for the next 6 weeks; the GEG group was fed a glutamine-containing diet for the first 2 weeks, and then a glutamine-containing diet with ethanol for the next 6 weeks.

**Table 7 nutrients-13-02788-t007:** Effects of glutamine supplementation on grip strength in chronic ethanol-fed rats ^1,2^.

Group	Grip Strength (g)		Difference in Grip Strength (g)	Difference in Grip Strength (g/kg BW)
	Initial	Final		
CC	1287.7 ± 206.8	1799.6 ± 254.8	511.9 ± 314.0	1217.9 ± 756.4
CE	1237.8 ± 135.2	1637.2 ± 282.1	399.4 ± 369.0	1021.0 ± 909.6
GG	1439.3 ± 137.2	1891.0 ± 127.0	451.7 ± 188.3	1115.1 ± 473.2
GE	1283.1 ± 146.6	1778.0 ± 240.2	494.9 ± 252.7	1306.3 ± 666.6
GEG	1438.7 ± 146.0	1766.0 ± 231.1	327.3 ± 339.3	867.0 ± 897.1

^1^ Values were expressed as the mean ± SD. ^2^ The CC group was fed the control diet for 8 weeks; the GG group was fed a glutamine-containing diet for 8 weeks; the CE group was fed the control diet the first 2 weeks and then an ethanol-containing diet for the next 6 weeks; the GE group was fed a glutamine-containing diet the first 2 weeks and then an ethanol-containing control diet for the next 6 weeks; the GEG group was fed a glutamine-containing diet for the first 2 weeks and then a glutamine-containing diet with ethanol for the next 6 weeks.

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
