# Peer review of "The Prophylactic Effects of Glutamine on Muscle Protein Synthesis and Degradation in Rats with Ethanol-Induced Liver Damage"

_nutrients, 2021, doi:10.3390/nu13082788_

Round 1

Reviewer 1 Report

The present paper entitled "Effects of Glutamine on Muscle protein Synthesis and Degradation in Rats with Etanol-Induced liver Damage" is a very interesting experimental work which demostrates that ethanol consumption in Wistar rats alters liver function, hepatic fatty change, elevates ammonia level , increases intestinal permeability and plasma endotoxin level and lowers levels of protein synthesis markers. The authors demostrated that all these ethanol induced injuries  are improved by glutamine supplementation. Moreover, a very elegant theory of  the complexicity of the mechanisms and pathways by which glutamine may have an positive effect on muscle protein synthesis is given in the figure at the end of the coclusions. I believe this paper is worth to be published. I have only one major and some minor comments.

Major comment:

  • The authors state that glutamine supplementation is ameliorating the effests of ethanol in rats with "chronic ethanol intake". A defininion of what "chronic ethanol intake" means is lacking. Are 6 weeks of ethanol feeding a long enough period to drive secure conclusions ?
  • On the other hand in the design of the experiment there were initially two groups of animals : those fed normally and those who had glutamine suplemmentation. After two weeks some of the animals were fed with ethanol. Based on this design of the experiment the effects of glutamine were rather prophylactic because the rats were first fed with glutamine and than with ethanol (after 2 weeks of glutamine). So the conclusion that glutamine is effective in chronic ethanol intake should maybe changed underlining " the prophylactic effect of glutamine". Otherwise a third group of animals first fed with classical diet together with ethanol for some weeks and than put on glutamine should be interesting to be included. This could be heplful in  better explaining the effects of glutamine in the case of chronic ethanol consumption. 

Minor comments

  • The definition of the 5 groups in the abstract is not clear, compared with the definition given in the text
  • Some comments about the limitations of the study in the discussion should be helpful (for example that only male rats were included in the study)

Reviewer 2 Report

Xiao Qian et al. analyzed the effect of glutamine intake in rats with alcohol induced liver injury. The researches found a beneficial effect of glutamine on kachexia and liver inflammation in this study. The study is of high quality and interest. There are some minor points which should be cleared by the researchers.

1.) Is it possible to provide the hepatic/plasma triglyceride levels?

2.) The fibrosis score would be of great interest. ist it possible to provide it?

3.) the discussion is full of hypothesis about potential pathways. None of this pathways was evaluated in this study. The discussion should be about the findings of this study and therefore should be modified.

4.) the English needs some editing.
